# *Berberis microphylla* G. Forst Intake Reduces the Cardiovascular Disease Plasmatic Markers Associated with a High-Fat Diet in a Mice Model

**DOI:** 10.3390/antiox12020304

**Published:** 2023-01-28

**Authors:** Lia Olivares-Caro, Daniela Nova-Baza, Claudia Radojkovic, Luis Bustamante, Daniel Duran, Daniela Mennickent, Victoria Melin, David Contreras, Andy J. Perez, Claudia Mardones

**Affiliations:** 1Departamento de Análisis Instrumental, Facultad de Farmacia, Universidad de Concepción, Concepción 4070386, Chile; 2Departamento de Bioquímica Clínica e Inmunología, Facultad de Farmacia, Universidad de Concepción, Concepción 4070386, Chile; 3Departamento de Ingeniería Mecánica, Facultad de Ingeniería, Universidad de Tarapacá, Avda. General Velásquez 1775, Arica 1000007, Chile; 4Departamento de Química Analítica e Inorgánica, Facultad de Ciencias Químicas, Universidad de Concepción, Concepción 4070386, Chile; 5Unidad de Desarrollo Tecnológico, Universidad de Concepción, Coronel 4191996, Chile

**Keywords:** *Berberis microphylla* G. Forst, calafate, polyphenols, cardiovascular disease risk, metabolomic

## Abstract

Polyphenols are bioactive substances that participate in the prevention of chronic illnesses. High content has been described in *Berberis microphylla* G. Forst (calafate), a wild berry extensively distributed in Chilean–Argentine Patagonia. We evaluated its beneficial effect through the study of mouse plasma metabolome changes after chronic consumption of this fruit. Characterized calafate extract was administered in water, for four months, to a group of mice fed with a high-fat diet and compared with a control diet. Metabolome changes were studied using UHPLC-DAD-QTOF-based untargeted metabolomics. The study was complemented by the analysis of protein biomarkers determined using Luminex technology, and quantification of OH radicals by electron paramagnetic resonance spectroscopy. Thirteen features were identified with a maximum annotation level-A, revealing an increase in succinic acid, activation of tricarboxylic acid and reduction of carnitine accumulation. Changes in plasma biomarkers were related to inflammation and cardiovascular disease, with changes in thrombomodulin (−24%), adiponectin (+68%), sE-selectin (−34%), sICAM-1 (−24%) and proMMP-9 (−31%) levels. The production of OH radicals in plasma was reduced after calafate intake (−17%), especially for the group fed with a high-fat diet. These changes could be associated with protection against atherosclerosis due to calafate consumption, which is discussed from a holistic and integrative point of view.

## 1. Introduction

Cardiovascular disease (CVD) is the main cause of death both in Chile and in the world: 27% of total deaths in Chile (MINSAL) and 31% worldwide (WHO) are caused by stroke or acute myocardial infarction. Hypertension, dyslipidemias, obesity, and metabolic syndromes, among other conditions, have been described as risk factors for CVD. Obesity is a condition characterized by an abnormal or excessive accumulation of fat, which leads to an altered metabolism and physiology [1]. A high-fat diet (HFD) in association with a sedentary lifestyle increases the probability of obesity [2]. A high-fat high-calorie diet has been related to increased oxidative stress, increased p47phox expression, reactive oxygen species (ROS) generation, NF-κB transcription, and a consequent elevation of TNF-α levels (a cytokine related to a proinflammatory state) [3]. In addition, Shai et al., 2006 described an imbalance of cytokines and inflammatory proteins in obesity and CVD. For instance, soluble forms of adhesion molecules, such as sICAM-1 and sVCAM-1, have been associated with obesity and other risk factors for coronary heart diseases (CHDs) [4]. Moreover, Ruel et al., 2008 demonstrated an association between oxidized LDL and sVCAM-1 plasma levels and an inverse correlation with sICAM-1 [5]. Furthermore, plasma E-selectin concentrations levels have been significantly correlated with hyperinsulinemia and insulin resistance assessed by HOMA-IR in men with abdominal obesity [6]. Leptin, a peptide hormone produced predominantly by white adipose cells, increases in obese individuals, and could promote inflammation related to chronic pathologies [7]. Finally, a review published by Bartekova et al., 2018 showed a large number of cytokines involved in heart disease: TNF-α, related to vascular dysfunction and atherogenesis; NF-kβ, associated with the expression of pro-inflammatory markers IL-2 and IL-4, which increase in plasma from stable angina pectoris patients; IL-6 increased in acute myocardial infarction; and IL-18 in oxidative stress, among others [8].

Polyphenols are natural antioxidants that reduce oxidation processes and inhibit free radical production [9]. Calafate is a barberry that is rich in polyphenols and that has a high antioxidant capacity obtained from an endemic shrub of Chilean–Argentine Patagonia [10,11,12]. It has been reported that calafate extract increased the ratio of reduced/oxidized glutathione (GSH/GSSG), which led to decreased oxidative stress in murine adipocytes treated with a macrophage-conditioned medium. This is considered an obesity in vitro model because obesity increases the macrophage population in white adipose tissue, with a proinflammatory phenotype [13]. Another study reported the inhibition of TNF-α gene expression by a calafate extract in murine macrophages stimulated with 5 μg/mL lipopolysaccharides. Finally, our research group recently described that calafate extract reduces intracellular ROS production (51%) in human umbilical vein endothelial cells (HUVECs) and completely inhibits LDL oxidation and malondialdehyde (MDA) formation, demonstrating a potential role for preventing oxidative stress and lipoperoxidation [14]. These studies suggest that calafate could contribute to reducing the proinflammatory and oxidative state induced by cardiovascular risk factors, such as HFD. However, in vivo studies are needed to evaluate the effect of calafate from an integrated biological response point of view.

Metabolomics is a tool that directly relates a measurable chemical response to a biological event, allowing a simultaneous semiquantitative comparison of hundreds to thousands of metabolites in a living system and linking the genotype and phenotype of an organism [15]. The metabolome represents a wide range of metabolites with different chemical and physical properties, which can be modified by internal and/or external factors [16]. Metabolomics compares patterns or “fingerprints” of metabolites that change in response to disease, exposure to toxins, and environmental or genetic alterations. Mass spectrometry (MS)-based metabolomics offers analyses with high selectivity and sensitivity and the potential to identify detected metabolites [15].

The main aim of this research was to evaluate the cardioprotective effect of calafate fruit through an in vivo nutritional intervention assay using a mice model fed a high-fat diet. Using LC–MS-based untargeted metabolomics, we evaluated the effect of chronic consumption of a calafate fruit extract on the metabolome of mice exposed to a high-fat diet. Additionally, we quantified the main inflammatory proteins and cardiovascular diseases, such as IL-6, insulin, leptin, MCP-1, PAI-1 total, resistin, TNF-α, E-selectin, ICAM-1, Pecam-1, P-selectin, proMMP-9, thrombomodulin and adiponectin, in plasma. Finally, we quantified OH radicals using electron paramagnetic resonance spectroscopy, a technique used in biology and medicine for the direct detection of radical species formed during oxidative stress. All these results were analyzed from an integrative point of view to obtain a holistic interpretation of the calafate extract effect in vivo.

## 2. Materials and Methods

### 2.1. Reagents and Vegetable Material

Formic acid (LiChropur^®^) for LC–MS (98–100%), acetonitrile (hyper grade), methanol (hyper grade), ethanol, water (HPLC–MS), and ammonium formate for mass spectrometry (>99.0%) were provided by Merck (Darmstadt, Germany). DL-2-aminoadipic acid, stearoyl-L-carnitine, propionyl-L-carnitine, trans-2-hexadecenoyl-L-carnitine, cis, cis-9,12-octadecadienoyl-L-carnitine, deoxycholic acid and succinic acid were obtained from Sigma–Aldrich (Steinheim, Germany).

Vegetable material: calafate berries (*Berberis microphylla* G. Forst) were collected near Punta Arenas, Chile (53.1548309, -70.911293).

### 2.2. Instrumentation

A Heraeus-Fresco17 centrifuge from Thermo Fisher (Waltham, MA, USA), an analytical balance from Denver Instrument Company (New York, NY, USA), and a refrigerated CentriVap concentrator from LABCONCO (Kansas City, MO, USA) were used for sample preparation.

The untargeted metabolomics was carried out with a UHPLC-DAD Bruker Elute LC system coupled in tandem with a Q-TOF spectrometer Compact, Bruker (Bremen, Germany). The control system used was Compass HyStar (Bruker), and the acquisition software was Bruker to control 4.1.402.322-7977-vc110 6.3.3.11. Data analysis was performed with Compass DataAnalysis 4.4.200 (Bruker) software, Compass QuantAnalysis 4.4 (Bruker Daltonics, Bremen, Germany), MetaboScape 3.0 (Bruker Daltonics, Bremen, Germany), and freely accessible software MetaboAnalyst (https://www.metaboanalyst.ca/, accessed on 15 December 2020).

Cytokine quantification was carried out in a MAGPIX instrument by Luminex XMAP Technology (Life Technologies, Grand Island, NY, USA) using a MILLIPLEX^®^ MAP mouse adipokine magnetic bead panel, MILLIPLEX^®^ MAP mouse adipokine magnetic single bead and MILLIPLEX^®^ MAP mouse CVD magnetic bead panel 1. Data analysis and data acquisition were carried out with Luminex xPONENT^®^ version 4.3229.0 (Luminex Corporation, TX, USA). Plate shaking for washing and incubation time was carried out in Thermomixer^®^ C serial number 5382KG639268 (Eppendorf, Hamburg, Germany).

#### 2.2.1. Calafate Extract

The extraction was carried out as previously described by Ruiz et al. [10], using ethanol/formic acid (97:3) as solvent extraction (raw extract), which was subsequently characterized. Anthocyanin was determined by HPLC-DAD by the method described by Ruiz et al. [10]. Hydroxycinnamic acid (HCA) and flavonol determination required a purification step using solid-phase extraction with cation-exchange columns (Oasis MCX Water, USA), as previously described by Ruiz et al. (purified extract) [11]. The raw extract was evaporated, posteriorly lyophilized and suspended in ultrapure water for the in vivo study in C57BL/67 mice (5 mg dry extract/mL). The calafate extract was characterized by HPLC-DAD-MS [17]. The sugar and ascorbic acid contents of the calafate extract were determined according to Morlock and Vega-Herrera, 2007 [18].

#### 2.2.2. In Vivo Assay

Ten- to twelve-week-old outbred C57BL/6J mice (n = 24) purchased from the Instituto de Salud Pública (ISP, Santiago, Chile) were housed in a conventional animal facility maintained at 25 ± 1 °C under a 12 h light:12 h dark photoperiod in accordance with the Guidelines for the Care and Use of Laboratory Animals (https://grants.nih.gov/grants/olaw/Guide-for-the-Care-and-use-of-laboratory-animals.pdf, accessed on 4 January 2023) [19]. After the acclimation period, mice were randomized and were fed ad libitum with a D12450H low-fat diet (10% fat; Open Source Diets-Research Diets, Inc., New Brunswick, NJ, USA), which is named a normal diet (N), or with a D12451 high-fat isocaloric diet (45% fat; Open Source Diets-Research Diets, Inc., New Brunswick, NJ, USA), which was named a high-fat diet (H) (n = 12 mice per group). After three months, each group was randomized into another two subgroups (n = 6 mice per group) that received either water (N and H groups) or calafate extract (5 mg dry extract/mL) in water, freshly prepared every 2 days, as the only source of liquid (called the Ncal and Hcal subgroups). Both calafate extract and water had similar pH values and sugar contents (pH 3 and 1.8 mg/mL of sugar) [17]. Pellet and liquid consumption were quantified daily, and animal weight was assessed weekly. At the end of the study, animals were fasted for 12 h at the beginning of the dark cycle, then were anesthetized with isoflurane and sacrificed through exsanguination at the fourth month after calafate extract supplementation. Blood samples were collected by cardiac puncture in EDTA tubes and centrifuged at 2500× *g* 15 min at 5 °C [17]. Aliquots of 20 μL plasma were frozen in liquid nitrogen and stored at −80 °C in Eppendorf tubes for metabolomic analysis.

#### 2.2.3. Biochemical Analyses and Protein Quantification

Cytokine quantification was performed using MAGPIX instruments with different immunological-based panels (MILLIPLEX MAP KIT) (See Section 2.2) for the following proteins: IL-6, insulin; leptin; MCP-1; total PAI-1 total; resistin and TNF-α; adiponectin; E-selectin; ICAM-1; Pecam-1; P-selectin; proMMP-9; and thrombomodulin. The assay was conducted following the manufacturer’s instructions using the MAGPIX system. Protein concentrations were calculated using the best-fitting parameter logistic on xPONENT^®^ software (Logistic 5P Weighted).

Biochemical analyses in plasma were carried out by colorimetric assay kits for plasma triglycerides (TGs) and total cholesterol (TC) (Valtek Diagnostics, Santiago, Chile). Alanine aminotransferase (ALT/GPT) and aspartate aminotransferase (AST/GOT) were measured using kinetic kits (Valtek Diagnostics, Chile). Blood glucose was measured using the OneTouch Ultra2 glucometer (Johnson & Johnson Medical). Measurements were performed in duplicate or triplicate according to the availability of samples using control sera (Valtrol N, Valtrol P; Valtek Diagnostics, Chile). The results are expressed as arithmetic means and standard deviations. Data were subjected to one-way ANOVA to evaluate the statistical significance of intergroup differences with the Levene test of homogeneity of variances and Bonferroni and Dunnett T3 post hoc tests, considering α < 0.05. Graphics were generated using the software Microsoft Excel for Microsoft 365 MSO version 2205, and statistical analyses were carried out using the software IBM SPSS version 20.

#### 2.2.4. Metabolite Extraction from Plasma Samples

Plasma samples were extracted for metabolomic analysis according to the protocol described below [20]. Briefly, 50 μL of plasma was mixed with 1000 μL of methanol (−20 °C). The samples were shaken for 1 min, stored for 5 min at 4 °C, and then vortexed for 5 s. Finally, the extracts were centrifuged at 15,700× *g* for 20 min at 4 °C. Then, 800 μL of each extract was transferred to a vial, and methanol was removed by vacuum centrifugal evaporation at 4 °C. Samples were reconstituted in 200 μL of methanol (−20 °C). A quality-control sample (QC) was obtained with 30 µL of each plasma sample. It was extracted as described previously and was injected between samples for robustness and repeatability evaluation of the instrumental system. The entire extraction procedure was performed in glass vials to avoid plastic contamination [21]. Methods showed a different number of samples because some were lost in the extraction procedure.

#### 2.2.5. UHPLC-DAD-ESI-QTOF-MS/MS Metabolomics Analysis

Plasma extracts were analyzed by reversed-phase chromatography UHPLC-DAD-ESI-QTOF-MS/MS based on the method previously described by Laursen et al., 2017 (Method 1) and Patterson et al., 2015 (Method 2) for compounds with medium polarity and apolar, respectively [22,23], with modifications. A Phenomenex column Kinetex^®^ C18 100 × 4.6 mm 2.6 μm (Torrance, CA, USA), oven temperature 50 °C, autosampler temperature 4 °C, and partial loop injection mode were used for both methods. Polar metabolites in plasma extracts were analyzed by normal phase chromatography based on the method described by Armirotti et al., 2014 for (Method 3) [24] with modifications. A HILIC BEH Amide 100 × 2.1 mm y de 1.7 μm (Waters, Milford, MA, EEUU), oven temperature 40 °C, autosampler temperature 4 °C, and partial loop injection mode were used. The details of the chromatographic conditions of these methods are described in Appendix A.

The MS conditions in all the metabolomic studies were positive ionization ESI +4500 V and negative ionization ESI −3500 V; dry gas, 9 L/min; nebulizer, 4 Bar; T°, 200 °C; end capillary 500 V; collision energy at 10–25 eV in stepping mode; Auto MS/MS mode (2 precursor/cycle), 50–1500 m/z (scan 0.2 s centroid mode); and internal calibration using sodium formate (0.01 M) with a mass accuracy < 3 ppm.

All metabolomic sequences started with 5 blank injections (solvent only), followed by 3 technical blanks (considering extraction process) and 3 QC. Then, every 11 samples, a QC was injected [25].

#### 2.2.6. Data Processing and Data Analysis

Bucket table: UHPLC-ESI-QTOF-MS/MS was processed in MetaboScape 3.0 (T-ReX 3D algorithm) under the parameters shown in Appendix A. This software transformed the raw data into a matrix with the features (m/z − retention time (tR) pairs and normalized peak intensity) (file: .csv). The workflow included mass recalibration, tR alignment, feature extraction (m/z − tR pairs), adduct and neutral loss administration, import of MS/MS spectra, and generation of the bucket table.

Data prefilter: The matrix was processed in MetaboAnalyst software, and the data were filtered considering RSDs > 20% compared to QC. The purpose of data filtering is to identify and remove variables that are unlikely to be of use when modeling the data. Principal component analysis (PCA) and volcano plot (fold change threshold: 1.5 and p value threshold: 0.05 FDR-adjusted) were used to select characteristic features of each group (comparisons between 2 group were: N-Ncal; H-Hcal; N-H and Ncal-Hcal). Peak area integration was manually curated with Compass QuantAnalysis 4.4 software.

Filter and feature selection: The new bucket table was processed in MetaboAnalyst 5.0, and data filtering and Pareto scaling were performed again. All variables were pareto-scaled to reduce the relative importance of large values and keep the data structure partially intact [1]. ANOVA (*p* < 0.05) was used to select the significant characteristic features of each group (Appendix A).

Metabolite annotation workflow: In DataAnalysis software, the molecular formula with the smallest mass error was selected. The spectra were sent to databases such as MetFrag (in silico fragmentation for computer-assisted identification of metabolite mass spectra) (https://msbi.ipb-halle.de/MetFrag/, accessed on 4 January 2023), HMDB (the human metabolome database) (https://hmdb.ca/, accessed on 4 January 2023), and METLIN (https://metlin.scripps.edu/landing_page.php?pgcontent=mainPage, accessed on 4 January 2023) by comparing the accurate mass and fragments information obtained from UHPLC-QTOF/MS [22,26]. Furthermore, the theoretical isotopic pattern was compared with the experimental isotopic pattern. Finally, identity confirmation of the most important metabolites was carried out by standard comparison (retention time (tR) and/or fragmentation spectrum). Metabolite annotation was based on the guide for annotation, quantification and best reporting practices proposed by Alseekh et al., 2021 [25]. Identification level (A; B; C) consisted of: (A) standard; (B), confident match based on MS/MS and confident match using in-silico MS/MS approaches and partial match based on MS/MS; (C) confident match based on MSn and C confident match using in-silico MSn approaches and partial match based on MSn; and (D) MS only. Tolerance to the m/z value was set to 10 ppm.

Heatmap analysis was carried out using Euclidean distances between samples to observe the group separation due to the differences in the metabolites identified in calafate consumption.

## 3. Results and Discussion

The effect of chronic consumption of calafate fruit extract on the plasma metabolome and its possible relationship with cardiovascular protection was evaluated by a metabolomic study of 24 plasma samples obtained from four groups of mice fed normal-fat and high-fat diets supplemented or not with calafate extract for 4 months (N, H, Ncal, Hcal) (see Section 2.2.2). The dose studied is equivalent to the intake of 210 g of fruit, which is within the recommended consumption of fruits and vegetables for the prevention of chronic diseases by the WHO. To reach this dose, the media extract intake by mice was 50 mg dry extract/day.

### 3.1. Calafate Berry Extract Composition

The calafate extract contained anthocyanins (225 ± 3 µmol/g dry weight), flavonols (12.0 ± 0.5 µmol/g dry weight) and HCA (28.4 ± 0.9 µmol/g dry weight). Major anthocyanins were delphinidin-3-glucoside (35%), petunidin-3-glucoside (19%) and malvidin-3-gluco-side (13%). Major flavonols and HCA identified were quercetin-3-rutinoside (36%) and quercetin-3-rhamnoside (27%), with caffeoylquinic acid (25%) and 3 or 4-trans-caffeoyl-glucaric acid (15%). The sugar, glucose and fructose contents in the extract were 64.3% *w*/*w*.

### 3.2. Animal Characteristics

According to the Guidelines for the Care and Use of Laboratory Animals (https://grants.nih.gov/grants/olaw/Guide-for-the-Care-and-use-of-laboratory-animals.pdf, accessed on 4 January 2023) [19], the mice showed no signs of stress. The average food and water intake per day in each group was 29.7 ± 6.8 g and 57.3 ± 7.7 mL, respectively. A significant increase in weight was observed for the group fed a high-fat diet H and Hcal (5.8%) (*p* < 0.05) (Appendix A) compared to initial weight. The average daily intake of calafate extract per group was 299.10 ± 51.94 mg of dry extract. The Hcal and Ncal groups did not show changes in body weight due to the consumption of calafate (Appendix A).

### 3.3. Biochemical Analysis

#### 3.3.1. Clinical Biochemistry

Biochemical analysis of plasma showed that total cholesterol (TC) was higher by 51% in the H groups than in the N groups (*p* < 0.05), which is similar to the results reported by Miao et al., 2016 and Guzmán and Sánchez, 2021 for high-fat diet rats [27,28]. No changes in TC were observed after chronic calafate intake; however, this cannot exclude modifications in lipoprotein levels of composition. Guzmán and Sánchez, 2021 reported an increase in the concentration of HDL and a decrease in the atherogenic index (−81%) and coronary risk index (−62%) posterior to calafate intake in a high-fat diet mouse model [28]. On the other hand, our results do not show significant differences for triglycerides, glycemia, and transaminase enzymes (GOT, GPT) (*p* > 0.05) (Appendix A).

#### 3.3.2. Inflammation and Cardiovascular Risk Markers

Concerning the plasma concentration of biomarkers associated with inflammation and cardiovascular disease, we observed significant changes, which are presented in Figure 1. The Ncal group showed a lower thrombomodulin concentration in plasma than the N group (*p* < 0.05), an effect that was not observed in the high-fat diet group. Thrombomodulin is a biomarker of vascular endothelial damage, whose increase has been associated with metabolic syndrome and CVD in children with risk factors for CVD [29]. Thrombomodulin plasma levels increase by approximately 43% in different groups of patients with atheromatous arterial disease (ischemic heart disease and polyvascular lesions) [30]. On the other hand, we observed an increase in adiponectin concentration in the Hcal (*p* < 0.05) and Ncal groups. Adiponectin is an endocrine factor mainly synthesized and released from adipose tissue. Adiponectin is a hormone secreted by adipocytes that is signaled through specific receptors in the skeletal muscle, vascular endothelium, liver, and myocardium. Studies have reported that adiponectin has insulin-sensitizing, vasodilator, antiatherogenic, and anti-inflammatory properties [31]. Okamoto et al., 2008, found that adiponectin-knockout mice developed 61% larger atherosclerotic lesions and accumulated 63% more CD4 T lymphocytes than control mice [32]. According to Masumi et al., 2011, adiponectin reduction in plasma markedly increases CHD risk in men and shows that the hazard ratio does not change after adjusting for BMI [33].

IL-6, insulin, leptin, MCP-1, PAI-1 total, resistin, TNF-α, sE-selectin, sICAM-1, Pecam-1, P-selectin and proMMP-9 did not show significant changes associated with calafate intake (Appendix A); however, sE-selectin, sICAM-1 and proMMP-9 showed a tendency to decrease in Hcal compared with those of the H group (Figure 1). Additionally, leptin decreased in the Ncal group compared to the N group, and this change was not observed in the H group (Figure 1). Adhesion molecule cell surface glycoproteins are expressed in endothelial cells and epithelial cells and have a role in epithelial injury-resolution responses, innate and adaptive immune responses in inflammation, and tumorigenesis [34]. Shai et al., 2006, showed that the soluble forms of adhesion molecules, such as sICAM-1 and sVCAM-1, are increased in plasma during endothelial injury and are useful as predictors of CHD [4]. High plasma levels of sICAM-1 and sVCAM-1 correlate with a 2.5-fold increase in CVD risk, and the sE-selectin concentration is increased in stroke [35]. In addition, these proteins were directly associated with body mass index (BMI), inflammatory biomarkers (IL-6, PCR, sTNF), and triglycerides and were inversely associated with high-density lipoprotein levels (*p* < 0.05) [4]. Cleavage of proMMP-9 results in the active enzyme MMP-9, which is a proteolytic enzyme (gelatinase) involved in the degradation of extracellular matrix, is secreted by various cells, including macrophages and endothelial cells, and contributes to atherosclerotic plaque susceptibility and rupture, leading to cardiovascular events such as myocardial infarction [36]. We observed a reduction tendency of proMMP-9 level in plasma for Hcal and Ncal, which can be associated to a minor level of MMP-9. The increase of proinflammatory cytokines are associated with synthesis of prom-MMP-9 [37]. Altogether, these results suggest that calafate-extract intake would have beneficial effects on cardiovascular disease.

Finally, leptin is an adipokine produced by white adipose tissue, and its disruption marks a milestone in the development of metabolic diseases, such as obesity, type 2 diabetes, and hypertension [38,39]. Agostinis-Sobrinho et al., 2017 showed a positive correlation between leptin and the leptin/adiponectin ratio and all metabolic risk factors (MRFs), such as triglycerides, HOMA-IR, and blood pressure. In addition, ROC-curve analyses showed that adiponectin, leptin, and the L/A ratio were predictive biomarkers of MRFs [40]. According to our results and based on these antecedents, regulation of these biomarkers produced by calafate-extract consumption can reduce the risk factors for CVD in mice.

### 3.4. Metabolomics

Typical base peak chromatograms of the plasma extract obtained with the different chromatographic methods are illustrated in Appendix A. After applying the described workflow (Section 2.2.6), the bucket table presented 39,227, 5464 and 15,778 features for methods 1, 2 and 3, respectively, in negative mode and 11,028, 19,929 and 11,070 in positive mode. After QC feature filtration [41], a total of 32,181 and 7652 features were deleted for negative and positive methods 1, 4030 and 16,657 for method 2, and 11,012 and 7844 for method 3. This supports the quality of the metabolomics data.

First, PCA examined spontaneous clustering patterns in the datasets, and as seen in Appendix A, the QCs were clustered in the center of the models. PCA models obtained with method 1 in negative ionization mode explained a total variance of 65.6% between principal component PC 1 and PC 2. In method 2, these components explained 80%, and method 3 explained 79.2%. In positive ionization, PCA showed values of 71.3% and 74.6% for methods 2 and 3, respectively. Method 1 was excluded because no significant features were obtained in the following analyses.

The cluster distribution of plasma samples denoted a separation between the N and H groups (Appendix A). Moreover, the PCA score plot of N-Ncal and H-Hcal (Figure 2) showed clustering of the samples due to calafate consumption. Relevant features were preselected by loadings plot and volcano plot analysis, and highly significant features were selected by ANOVA (*p* < 0.05). The significant features in the negative mode were 17, 11, and 37 for methods 1, 2, and 3, respectively, and in positive mode 7 and 44 for methods 2 and 3 (Appendix A).

#### Metabolite Annotation and Observational Changes

Metabolite annotation was carried out using the HMDB, METFRAG, METLIN, and PubChem databases and/or by comparison with the standards. The time retention, pseudomolecular ion, and fragments of significantly annotated plasma metabolites, related to cardiovascular disease, are presented in Table 1.

DL-2-Aminoadipic acid (2AAA) (N°2: [M-H]- 160.0612 m/z and N°14: [M+H]+ 162.0758 m/z) was also identified in a negative and positive mode in the HILIC method. 2-AAA was increased in the H group compared to the N group, and calafate-extract intake reduced its concentration in the H group and N group (*p* < 0.05) (Figure 3). 2-AAA is an oxidized derivative from the amino acid lysine. This metabolite has been strongly associated with diabetes development in the Framingham Heart Study (risk factor for CVD), where individuals with high plasmatic 2-AAA showed a 4-fold higher probability of developing diabetes after a 12-year follow-up period than those with the lowest concentration [46]. In addition, Estaras et al., 2020, demonstrated that the incubation of pancreatic acinar cells with 2-AAA induced oxidative stress and lipid peroxidation, both processes related to diabetes and CVD [47]. Finally, this metabolite has also been related to obesity, where 2-AAA levels are higher in the obese group than in the normal-weight group [48]. Plasma 2-AAA was positively associated with adiposity indices (fat mass, fat percent, waist circumference, BMI, and BMI z score; all *p*  ≤  0.0336) [48]. In the same study, researchers reported that 2-AAA levels increased in the adipose tissue of mice fed a high-fat diet compared with mice fed a standard diet, which is concordant with our results.

An upregulation in deoxycholic acid (N°1: [M-H]- 391.2849 m/z) was found in Group H, and this effect was reduced by calafate intake (Hcal), reaching a similar level found in the normal diet groups (*p* < 0.05) (Figure 3). This metabolite was detected by two chromatographic methods (1 and 2) in negative mode. Deoxycholic acid is a secondary bile acid generated by the 7α-dehydroxylation of cholic acid in gut microbiota [42]. Shimizu et al., 2014, demonstrated that this metabolite (5 µM) incubated for 48 h with vascular smooth muscle cells (VSMCs) promoted cellular migration and proliferation, related to endothelial dysfunction key in CVD [43]. Additionally, Haeusler et al., 2013, reported that plasma 12α-hydroxylated biliary acids are increased in association with insulin resistance, a risk factor for CVD [44]. A metabolomic study in the plasma of humans (60 control and 40 diabetic volunteers) showed deoxycholate or cholanoic acid in 68% of diabetic patients and 45% of control volunteers [45]. Finally, the increase in deoxycholic acid in the serum of high-fat-diet-fed mice analyzed by liquid chromatography–mass spectrometry has been previously reported [42], which is concordant with our findings.

Stearoyl-L-carnitine (N°9: [M+H]+ 428.3702 m/z) and linoleoyl-L-carnitine (N°11: [M+H]+ 424.3407 m/z) were identified in positive mode [49,50]. A tendency to decrease was observed from the H to the Ncal group (Figure 3). Significant differences were found between the N–H group and the Ncal–Hcal group (*p* < 0.05). Similar results have been reported by Mihalik et al., 2010, who described an increase in acylcarnitines in the plasma of obese volunteers and participants with type 2 diabetes mellitus, suggesting a defect in the use of succinyl-CoA in the tricarboxylic acid cycle and an increase in incomplete β-oxidation in skeletal muscle [51]. On the other hand, a metabolomic study of Langerhans islets (LH) in mice and humans with type 2 diabetes mellitus described acylcarnitine accumulation in LH, which could result from excessive β-oxidation in the presence of abundant fatty acids [52], which is consistent with our results. This study also noted specific stearoylcarnitine and linoleoylcarnitine accumulation and impaired tricarboxylic acid (TCA) cycle in mitochondrial energy metabolism due to a decrease in succinate and ATP [52]. Our results showed that calafate-extract intake diminished the increased stereoyl-L-carnitine and linoleoyl-L-carnitine observed in the H and N groups (Figure 3); these results, together with the increased succinate (N°3 117.0192 m/z) and propionyl-L-carnitine (N°13: [M+H]+ 218.1388 m/z) concentrations in the H group (H-Hcal *p* < 0.05) (Figure 3) [49,53], suggest a beneficial effect on oxidative phosphorylation and the TCA cycle, which did not cause an increase in oxidative stress. In the same way, using electron paramagnetic resonance (EPR) spectroscopy based on Tarifeño-Saldivia, 2018 [54], we observed a hydroxyl radical reduction of 17% in Hcal plasma compared with H (*p* < 0.05), which demonstrated the antioxidant effect of calafate on plasma (Appendix A). Propionyl-L-carnitine has high affinity for carnitine acyltransferase, producing propionyl-coenzyme A and L-carnitine, essential cofactors in the transport of long-chain fatty acids from the cytosol to the mitochondria [55]. On the other hand, treatment with propionyl-L-carnitine decreased lipid peroxidation (malondialdehyde formation) in SHRs (hypertensive mice) by 40% in the liver and 34% in the heart [56]. The same tendency to decrease from H to Ncal was not found for trans-2-hexadecenoyl-L-carnitine (N° 8: [M+H]+ 398.3251 m/z), although it is from the same family of compounds.

Considering the abnormal metabolism of fatty acids found in the H group, we also found a significant increase in tetracosahexaenoic acid in the H group (N° 12: [M+H]+ 357.2767 m/z) (*p* < 0.05), similar to the report of Miao et al., 2016 (Figure 3) [27]. The group that received the calafate extract (Hcal) showed a reduction in this metabolite, supporting the improvement of β-oxidation.

These findings suggest that the intake of calafate favors β-oxidation and may have an FFA-lowering effect in prolonged consumption. Moreover, calafate intake reduced metabolites associated with inflammation and CVD, which supports its beneficial effects on health. These results can also be observed in the heatmap analysis (Figure 4), where the H group is totally separated from Hcal, N and Ncal, highlighting a tendency of the Hcal group to be closer to the N group. These metabolites are associated with a high-fat diet and are modified by calafate consumption.

Additionally, it is worth noting that metabolites with no significant differences between paired-group comparisons (H-Hcal and N-Ncal) showed differences between the normal and high-fat diets (Figure 3). These metabolites were Lyso PC (16:1/0:0) (N°4: [M+FA-H]- 538.3140 m/z) and Lyso PC (20:5) (N°5: [M+FA-H]- 586.3142 m/z), which were tentatively identified in negative mode, and Lyso PC (18:4) (N°10: [M+H]+ 516.3046), which was detected in positive mode. Ganna et al., 2014, associated Lyso PC (18:2) and (18:1) with high HDL-C and total cholesterol levels and with low BMI and subclinical CVD markers. They also found a strong negative association between LysoPC and coronary heart disease [57]. Additionally, an atherogenic diet decreased Paraoxonase-1 ARNm and activity, an enzyme responsible for producing LysoPC from phosphatidylcholine (PC), which is capable of limiting the synthesis of cholesterol in macrophages with antiatherogenic effects [58,59].

In addition, we identified Lyso PE (16:1) (N°6: [M-H]- 450.2619 m/z) [60] and Lyso PE (20:4) (N°7: [M-H]- 500.2775 m/z) (Figure 3). The first was higher in the N and Ncal groups than in the H and Hcal groups (*p* < 0.05). However, LysoPE (20:4) showed significant differences between the N and Ncal groups (*p* < 0.05) and between the H and Hcal groups (*p* < 0.1). Lyso PEs containing unsaturated fatty acids, which have been related to coronary artery disease (CAD), have higher levels in patients with CAD than in control subjects [61]. In this context, these differences are associated with the high-fat diet model.

Altogether, these results demonstrate that high-fat-fed mice display metabolomic changes that suggest an impairment of β-oxidation and mitochondrial energy related to cardiovascular disease development. Calafate consumption can modify this profile, reducing the impact of a high-fat diet on metabolism.

### 3.5. Biological Interpretation

Phenol compounds are the main compounds in calafate berries and have been reported to have effects such as ROS scavenging, induction of enzymes that scavenge ROS and synthesize endogenous antioxidants, metal chelation, inhibition of ROS-producing enzymes such as NADPH oxidase, and effects on the electron transport chain, among others [62]. Anthocyanins (the main polyphenols in calafate) can trap free radicals, reducing oxidative stress. Delphinidin is able to protect SOD activity in HUVECs and attenuate ox-LDL-induced generation of ROS, p38MAPK protein expression, NF-kβ activity and protein expression, IκB-α degradation and expression of adhesion molecules (P-selectin and ICAM-1) in endothelial cells [63]. Flavonols such as quercetin decrease the levels of MMP-9 and NF-kβ in mice fed 0.2% quercetin-fortified rodent chow [64], and significantly increase the oxygen consumption rate and energy metabolism (glycolysis and mitochondrial respiration) in AML12 hepatocytes, suggesting enhanced fatty acid β-oxidation. It has also been demonstrated that quercetin affects the expression levels of lipid metabolism-related genes (Ppara and Pparg) [65]. Finally, chlorogenic acid (the main hydroxycinnamic acid from calafate) and berberine (alkaloid in calafate) have been reported to increase adiponectin levels in visceral adipose tissue and serum, respectively [31]. Adiponectin drastically increases the expression and activity of PPAR-α, promoting fatty acid oxidation and tricarboxylic cycle activation in the liver. Adiponectin inhibits NF-kβ activation, mediating vascular cell adhesion molecule-1 expression in endothelial cells [31]. Peroxisomal beta-oxidation can also be affected by the increment of adiponectin observed after calafate intake. However transcription factors as PPAR-a must be determined. This factor is related to very- long-chain fatty acid oxidation in the peroxisomal organelle, in which are shortened through a specific β-oxidation system. Then, shortened fatty acids are metabolized by mitochondrial β-oxidation reducing the toxicity of this long chain and branched-chain fatty acid in liver [66]. These findings did not consider the antagonistic and synergistic effects that could be present in calafate, so studies with the complete composition of fruit are necessary.

Previously, our research group reported that calafate extract reduced oxidative stress by reducing intracellular ROS production (51%) and completely inhibiting LDL oxidation, which is attributed to its complex composition of bioactive compounds [14]. Calfío and Huidobro-Toro, 2019, described a vasodilator effect of calafate-berry extract through nitric oxide, one of the most important endothelium-derived vasodilator molecules [67]. Reyes-Farias et al., 2016, demonstrated that calafate-berry extract increased the ratio of reduced/oxidized glutathione (GSH/GSSG), which led to decreased oxidative stress in murine adipocytes treated with macrophage-conditioned medium [13]. Finally, in this work, we demonstrated that calafate extract decreases endothelial dysfunction and inflammatory markers, acylcarnitine derivatives and metabolites associated with cardiovascular disease. Additionally, an increase in succinic acid is related to TCA activation and an increase in adiponectin hormone, which have been shown to have pleiotropic effects.

In accordance with the results described previously, we propose in Figure 5 that calafate could have an effect on three main organs: adipose tissue, liver, and vascular endothelium. Calafate fruit decreases oxidative stress and can activate transcription factors, such as NF-kβ and PPAR-α, decreasing vascular adhesion molecules and activating fatty acid oxidation and the TCA cycle in the liver. On the other hand, this fruit could stimulate an increase in adiponectin levels in adipose tissue. Transcriptional studies in in vivo models should be carried out to corroborate this new hypothesis.

## 4. Conclusions

The current study presents for the first time the beneficial impact of ad libitum feeding of calafate-berry extract on the plasmatic proteins and metabolome in mice exposed to a high-fat diet. Calafate intake produced a change in plasma proteins and the metabolome of mice fed a HFD, both related to a reduced risk of CVD associated with the HFD. A reduction in thrombomodulin, sE-selectin, sICAM-1 and proMMP-9, which are associated with endothelial dysfunction as bases of atherosclerosis, was observed. Additionally, a reduction in leptin and an increase in adiponectin were found, both positively associated with the reduction in CDV risk. Significant changes in the metabolome could explain the role of calafate in the β-oxidation of fatty acids. These changes include an increase in succinic acid levels, activating tricarboxylic acid and reducing carnitine accumulation. Additionally, other metabolites associated with endothelial dysfunction, oxidative stress and lipid peroxidation, such as deoxycholic and aminoadipic acid, were also detected. All these findings can explain an effective role of calafate consumption in reducing the risk of cardiovascular disease caused by a high-fat diet.

## Figures and Tables

**Figure 1 antioxidants-12-00304-f001:**
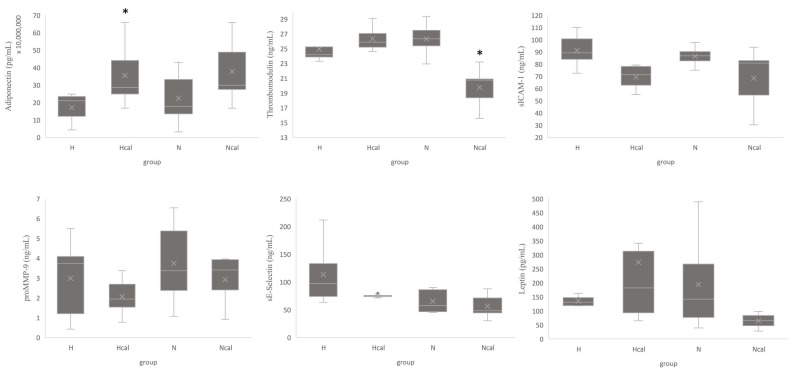
Box-plot proteins markers in endothelial dysfunction and adipokines related with metabolic risk factors. Data expressed as concentration in plasma samples (ng/mL or pg/mL) in different groups (mean X, median line and interquartile range in box-plot, * *p* < 0.05 H-Hcal N-Ncal). H—High-fat diet group; N—normal diet group; Hcal—High-fat diet group with calafate; Ncal—Normal diet with calafate.

**Figure 2 antioxidants-12-00304-f002:**
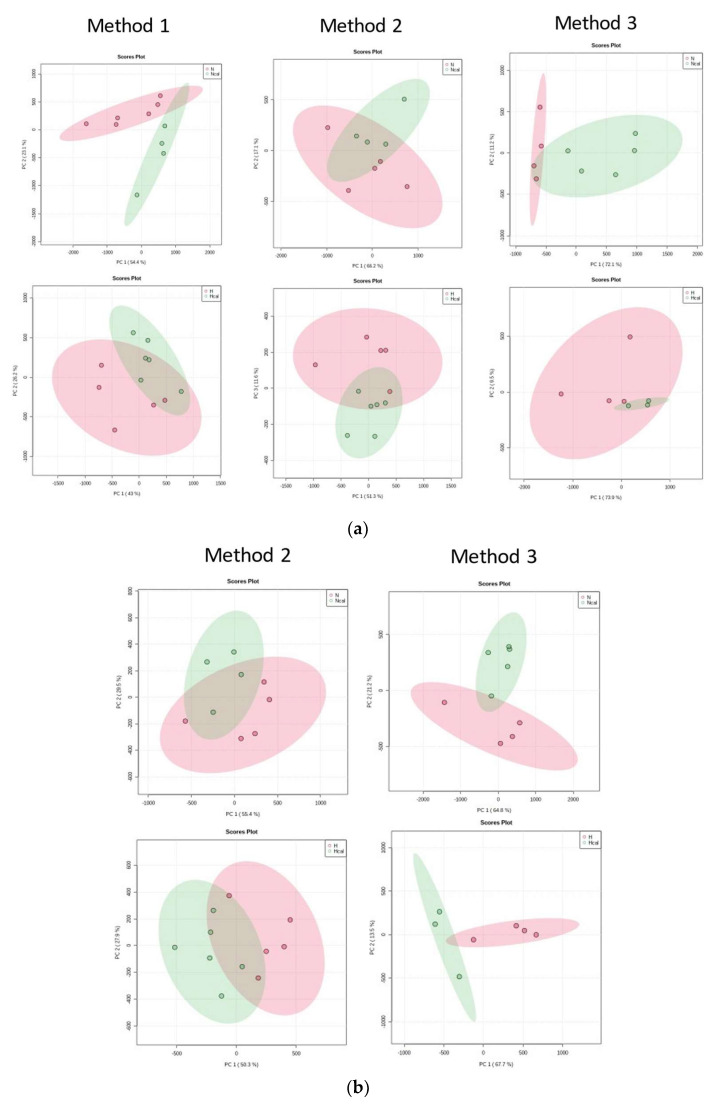
Principal component analysis (PCA). (**a**) Score plot in negative mode and (**b**) Score plot in positive mode for method 1, method 2 and method 3. N and H in red, and Ncal and Hcal in green for each ionization mode. H—High-fat diet group; N—normal diet group; Hcal—High-fat diet group with calafate; Ncal—Normal diet with calafate.

**Figure 3 antioxidants-12-00304-f003:**
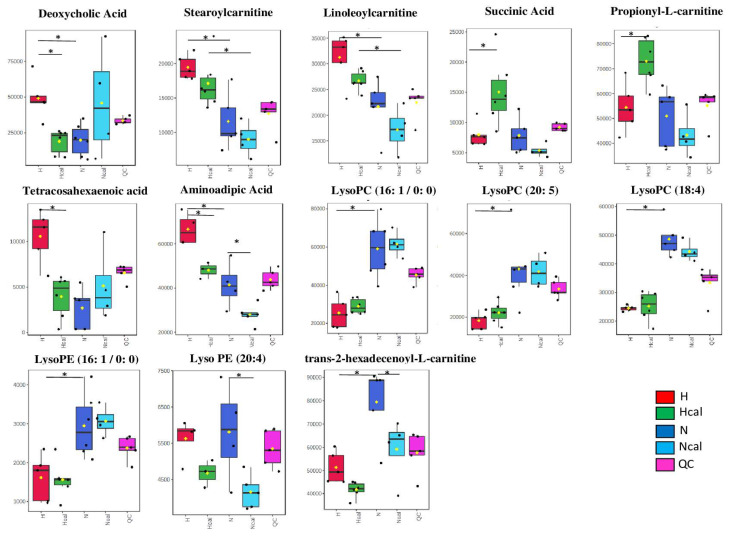
Box plot features significantly (* *p* < 0.05) identified. H in red, Hcal in green, N in blue, Ncal in light blue and QC in pink. H—High-fat diet group; N—normal diet group; Hcal—High-fat diet group with calafate; Ncal—Normal diet with calafate; QC—Quality control. Y-axis corresponds to the peak area.

**Figure 4 antioxidants-12-00304-f004:**
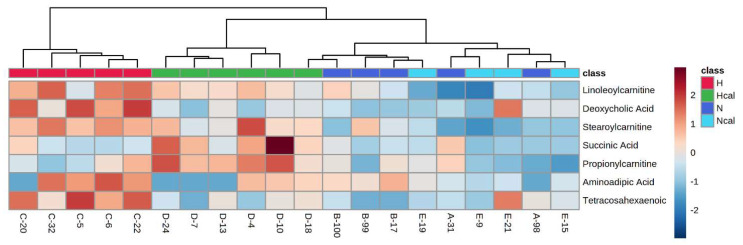
Heatmap of significant metabolites. H—High-fat diet group; N—normal diet group; Hcal—High-fat diet group with calafate; Ncal—Normal diet with calafate. C-20 to E-15 are the labels of each analyzed sample.

**Figure 5 antioxidants-12-00304-f005:**
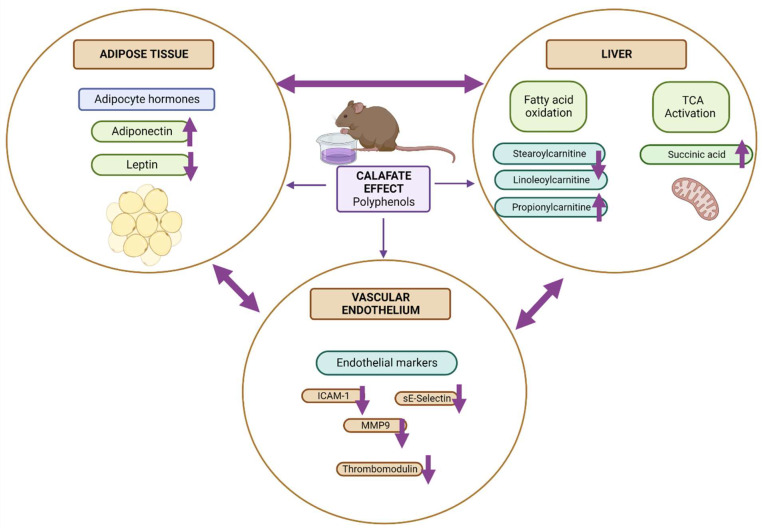
Proposal of the systemic effect caused by *Berberis microphylla* G. Forst (calafate) berries.

**Table 1 antioxidants-12-00304-t001:** Significant metabolites identified by UHPLC-DAD-QTOF which are related with CVD (positive and negative ionization modes).

Feature N°	Ion	Molecular Ion Formula	m/z	Fragments	Error ppm	Identification	Identification Level (A-D) ^1^	International Identifier ^2^	Method ^3^
	[M-H]-	[C_24_H_39_O_4_]-	391.2849	391.2872; 345.2823 C_23_H_37_O_2;_ 392.2978	−1.9	deoxycholic acid	A	METLIN 265	1
1								HMDB00626	
2	[M-H]-	[C_6_H_10_NO_4_]-	160.0612	116.0709 C_5_NO_2_H_10;_ 142.0510 C_6_H_8_NO_3_	3.5	DL- 2-aminoadipic acid	A	METLIN 324 HMDB00510	3
3	[M-H]-	[C_4_H_5_O_4_]-	117.0192		−2.4	succinic acid	A tR	METLIN 114	2 3
4	[M+FA-H]-	[C_25_H_49_NO_9_P]-	538.3140	253.2184 palmitoleic acid C_16_H_29_O_2;_ 478.2965 C_23_H_45_NO_7_P; 479.3015; 254.2202; 224.0688 C_7_H_15_NO_5_P; 538.3203; 152.9954 C_6_H_4_NO_2_; 242.0767 C_7_H_17_NO_6_P; 168.0431 C_4_H_11_NO_4_P; 255.2270 palmitic acid C_16_H_31_O_2;_ 480.3119; 134.9973 82 C_7_H_4_OP	−2.5	LysoPC(16:1/0:0)	B	HMDB0010383	1, 2 and 3
								METLIN 40288	
5	[M+FA-H]-	[ C_29_H_49_NO_9_P ]-	586.3142	526.2964 C_27_H_45_NO_7_P; 527.2954; 301.2199; 224.0732; 586.3142; 587.3137; 303.2374 C_16_H_34_NO_2_P; 302.2256; 152.9942; 528.2938; 301.2686 C_20_H_29_O_2;_ 529.3097; 168.0459; 228.2245; 257.2409 C_16_H_34_P; 242.0797 C_7_H_2_NO_6_P; 257.0238 C_13_H_8_NO_3_P	−0.9	LysoPC (20: 5 / 0: 0)	B	HMDB10397	1
6	[M-H]-	[ C_21_H_41_NO_7_P]-	450.2619	253.2162 palmitoleic acid C_16_H_29_O_2;_ 450.2641; 254.2229; 451.2742; 196.0359 C_5_H_11_NO_5_P; 121.9994 C_2_H_5_NO_3_P; 197.0393; 140.0115 C_2_H_7_NO_4_P; 262.9691; 451.3470	4.2	LysoPE (16:1/0:0)	B	HMDB0011504	1
						LysoPE (0:0/16:1)		HMDB0011474	
7	[M-H]-	[C_25_H_43_NO_7_P]-	500.2775		−2.9	LysoPE(20:4)	D	METLIN 62302, 62275 or 62276	3
8	[M+H]+	[C_23_H_43_NO_4_]+	398.3251	85.0260 182 C_5_H_4_O_2_; 125.0717; 144.1003 C_7_H_14_NO_2_	−1.9	trans-2-hexadecenoyl-L-carnitine	A	METLIN 58388	2
								HMDB06317	
9	[M+H]+	[C_25_H_50_NO_4_]+	428.3702		9.1	stereoyl-L-carnitine	A (tR)	METLIN 5811	2
10	[M+H]+	[C_26_H_47_NO_7_P]+	516.3046	184.0733 C_5_H_15_NO_4_P; 124.9998; 98.9853; 104.1061; 457.2327; 458.2310; 86.0967; 185.0763 C_5_H_12_N; 146.9791; 311.2591	7	LysoPC (18:4)	B (family Standard D)	HMDB10389	2
								METLIN 61699	
11	[M+H]+	[C_25_H_46_NO_4_]+	424.3407	97.1010	2.1	cys,cys-9,12-octadecadienoyl-L-carnitine	A	METLIN 58418	2 and 3
						linoleoyl-L-carnitine			
12	[M+H]+	[C_24_H_37_O_2_]+	357.2767		−3.7	tetracosahexaenoic acid	D	METLIN 6430	2
13	[M+H]+	[C_10_H_20_NO_4_]+	218.1388		3.8	propionyl-L-carnitine	A (tR)	METLIN 965	2 and 3
14	[M+H]+	[C_6_H_12_NO_4_]+	162.0758	98.0587 C_5_H_8_NO	1.7	aminoadipic acid	B	HMDB00510	3
								METLIN 324	

^1^ Identification level based in Alseekh et al., 2021 [25]. ^2^ Data base used for identification. ^3^ Method of analysis.

## Data Availability

Not applicable.

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
