# Peer review of "Berberis microphylla G. Forst Intake Reduces the Cardiovascular Disease Plasmatic Markers Associated with a High-Fat Diet in a Mice Model"

_antioxidants, 2023, doi:10.3390/antiox12020304_

Round 1
Reviewer 1 Report
Manuscript presented to review with title: “Berberis microphylla G. Forst Intake Reduces the Cardiovascular Disease Plasmatic Markers Associated to a High Fat Diet in a Mice Model” is very interesting and well written.
Presented manuscript is on good scientific level and represent a very high scientific value manuscript.
The Abstract. Authors give a short presentation of manuscript. This section is well constructed. Please add some representative values for main results to Summary section.
Introduction section.
The Introduction section includes all necessary information about examined objects and problems. Only one comment to this part of manuscript:
Formatted aim of the work reflect more “What we did” not typical aim. Could you Change it a little bit to be more similar to the aim of the work?
Materials and method section are presented all necessary information. In my opinion, this section is correct.
Results.
Presented in manuscript tables and figures are good constructed. They are good for reading and understanding.
The discussion section presents a good comparison of the obtained results with other results available in the data basis.
Presented conclusions are corresponding with all information presented via Authors’ in manuscript text.
General opinion: After carefully manuscript reading, I think, that presented manuscript is a very valuable. In my opinion Manuscript should be accept with some small correction according to my suggestions and after that publish in Antioxidants journal.
Reviewer 2 Report
This manuscript describes that the chronic intake of Berberis microphylla G. reduces the cardiovascular risk associated to a high fat diet. A large amount of inflammatory factors and proteins related to cardiovascular diseases are supposedly reported, including IL-6, insulin, leptin, MCP-1, PAI-1, resistin, TNF-α, E-selectin, ICAM-1, Pecam-1, P-selectin, proMMP-9, thrombomodulin and adiponectin, in plasma of model mice fed with a water extract of the south american berry above mentioned (see introduction). However,
But only six are showed at Figure 1. The first modification should be related to this point. Abstract is correct, but the introduction should be modified (lines 87-89 we investigated …… as some of these biomarkers are not actually investigated).
Aside this point, the introduction is correct, focused to the goal of the manuscript and brief enough concerning metabolomic, techniques used for the study and main antioxidant components of Berberis microphylla G.
Concerning methods for UHPLC-DAD-ESI-QTOF-MS/MS metabolomics analysis. Methods for HPLC analysis concerning solvents, gradients and so on are well described. Same for MS conditions, but the discussion of the 3 methods is not really necessary. Provided there are supplementary material, some of this information could be translated, and only methods with higher significance must remain in the regular paper. Authors might re-think about that.
On the other hand, the reliability of the identified biomolecules is not warranted, and some of them are not usual metabolites in spite of the useful servers used for identification. For example, eicosapentaenoic acid, trans-2-hexadecenoyl-L-carnitine and
tetracosahexaenoic acid. There are several reasons to doubt about those identifications.
The levels of tissue inhibitors of metalloproteinases TIMP are not measured, so that proMMP9 and MMP9 could have different pattern. The manuscript refers sometimes to proMMP9 and sometimes MMP9, but this is not clear enough. Please clarify whether the analytical determination is MMP9 activity or proMMP9 protein
Other minor points
Line 265: Are 210 g of fruits the total ingestion per animal in 3 months?. How much is that dairy or weekly?, The concentration in the water extract would be convenient (line 145).
Anthocyanins are the main antioxidant components of the berries. So, these compounds should be mentioned at methods, as in line 126 and more important at line 572. “Flavonols such as quercetin decrease the levels of MMP-9”
Line 595: propyl and stearylcarnitine are acylcarnitine derivatives. So, the sentence should be expressed more precisely avoid the general term acylcarnitine.
Figure 3: Define y-axis scale.
Figure 4: The meaning of the x-axis labels should be defined (from C-20 to E.15).
Figure 5: This proposal is not totally consistent with the obtained results. Thrombomodulin is not at the proposal, and this is one of the clearest data obtained
Liver: Linoelaidylcarnitine? How confidence is that identification. This is also a trans isomer. This metabolite is not found in results, so it cannot be in the proposal. On the other hand, trans-hexadecenoyl is in results. This is c16, but linoelaidyl in the proposal is C18. Check it please
Line 560: impairment of β-oxidation and mitochondrial energy line 601. The peroxisomal beta-oxidation should be mentioned, mostly after PPAR-α comment at line 601. Peroxisomal beta-oxidation should be taken into account for long FA, such as the C24. The role of carnitine in mitochondria and peroxisomes is different and this should be bear in mind.
Round 2
Reviewer 2 Report
The point-by-point reply letter is clear and as much convincing as possible concerning the identification of some metabolites. Metabolomic data base have been used with expertise that the authors reply satisfy my concerns. Other points are rightly addressed. Just delete m at line 320 "prom-MMP-9 [30]". Good job.